# Facile Synthesis of the Polyaniline@Waste Cellulosic Nanocomposite for the Efficient Decontamination of Copper(II) and Phenol from Wastewater

**DOI:** 10.3390/nano13061014

**Published:** 2023-03-11

**Authors:** Ahmed N. Doyo, Rajeev Kumar, Mohamed A. Barakat

**Affiliations:** Department of Environmental Sciences, Faculty of Meteorology, Environment and Arid Land Agriculture, King Abdulaziz University, Jeddah 21589, Saudi Arabia

**Keywords:** PANI@WTP nanocomposite, adsorption, copper, phenol, kinetics, isotherms

## Abstract

The existence of heavy metals and organic pollutants in wastewater is a threat to the ecosystem and a challenge for researchers to remove using common technology. Herein, a facile one-step in situ oxidative polymerization synthesis method has been used to fabricate polyaniline@waste cellulosic nanocomposite adsornt, polyaniline-embedded waste tissue paper (PANI@WTP) to remove copper(II) and phenol from the aqueous solution. The structural and surface properties of the synthesized materials were examined by XRD, FTIR, TEM, and a zeta potential analyzer. The scavenging of the Cu(II) and phenol onto the prepared materials was investigated as a function of interaction time, pollutant concentration, and solution pH. Advanced kinetics and isotherms modeling is used to explore the Cu(II) ion and phenol adsorption mechanisms. The synthesized PANI@WTP adsorbent showed a high intake capacity for Cu(II) than phenol, with the maximum calculated adsorption capacity of 605.20 and 501.23 mg g^−1^, respectively. The Langmuir equilibrium isotherm model is well-fitted for Cu(II) and phenol adsorption onto the PANI@WTP. The superior scavenging capability of the PANI@WTP for Cu(II) and phenol could be explained based on the host–guest interaction forces and large active sites. Moreover, the efficiency of the PANI@WTP for Cu(II) and phenol scavenging was excellent even after the five cycles of regeneration.

## 1. Introduction

Wastewater containing pollutants such as heavy metals and organic compounds reduces water quality and threatens the ecosystem and human health. Removing and recovering organic contaminants and metal ions from aqueous effluents is difficult. This is due to their varied physical and chemical characteristics, which hinder the complete removal and recovery. Industrial and mining activity and other sources introduce heavy metals and organic pollutants into water systems [1]. Water containing harmful metals such as copper, iron, lead, mercury etc., and organic debris such as phenol has health-related consequences for humans and animals. Copper is a micronutrient trace element required for human and animal nutrition. Copper is toxic and carcinogenic and can accumulate in the liver, leading to stomach cramps, breathing problems, and liver and kidney loss [2]. On the other hand, phenol and its derivatives are highly toxic compounds that significantly impact human health and aquatic species. Approximately 6 million tons of phenol are generated each year globally, with a marked upward tendency [3,4].

Among the technologies reported to remove hazardous heavy metal cations and phenolic compounds from polluted water are chemical precipitation [5], membrane processes [6], flotation [7], solvent extraction [8], oxidation [9], adsorption [10], ion exchange [11], and electrochemical methods [12]. Each technology has some advantages and disadvantages. However, most of these treatment methods have shortcomings such as expense, secondary pollution, or being poor/ineffective etc. For instance, chemical precipitation generates more sludge since it uses more chemicals than necessary to treat the water, while other technologies need high capital investment and energy. However, the adsorption technique is considered relatively cost effective, practical, and simple to use [5,13,14]. The applications of convectional adsorbent materials, such as activated carbon (AC) [15], zeolite [16,17], and resins [18], and so forth, are widely investigated to decontaminate organic as well as inorganic pollutants from water streams. Scientists have recently been searching for the fabrication of low-cost adsorbents, mainly made from biopolymers such as alginate, cellulose, waste biomass, etc., as they are low-cost, abundantly available, and efficient adsorbents [19].

Cellulose (C_6_H_11_O_5_) is the most abundant organic biopolymer on Earth [20,21]. Cellulose can be used to produce a variety of nanomaterials, cellulose nanofibrils, and oxidized cellulose nanoparticles (CNFs) etc. Organic moieties such as carboxylic and hydroxyl groups can be changed or grafted with other functional groups to create new functional groups that can be adapted explicitly for application. These functional groups ensure that the addition of nanoparticles can modify cellulose biopolymers. The cellulose materials’ functional groups are crucial to the adsorption of metal ions [22]. Chemically modified cellulose has a greater adsorption capacity for various aquatic contaminants than its unmodified counterparts. Organic compounds, acids, bases, minerals, oxidizing agents, and other substances have been employed to modify cellulose [23,24]. Tissue paper (TP) is a cellulose fiber-rich material widely used for multiple purposes. It produces a massive amount of waste after its use. Waste tissue paper (WTP) has been used to prepare carbon aerogel and activated carbon (AC), and is utilized as an ideal candidate for the adsorption of organic chemicals and heavy metals from effluent. Products made of spongy tissue absorb water quickly while maintaining a high absorption capacity throughout the task. The ability to absorb liquid relies on enough surface tension to suck out the liquid and a high permeability. Due to the high hydrophilicity, TP absorbs a large amount of water and fewer pollutants. Therefore, to enhance pollutant adsorption and reduce its hydrophilicity, TP should be altered with low surface energy materials [25].

Among conductive polymers, polyaniline has gained huge research attention because of its important characteristics, including a simple synthesis route, environmental stability, good electrical conductivity, etc. Various applications related to active functional groups, such as amine, imine, and secondary amino groups on the polyaniline structure, make it suitable for removing dyes, toxic metals, and organic chemicals and dyes from the aqueous phase [26]. However, PANI’s significance is constrained by several drawbacks, including insolubility or low/partial solubility in common solvents, long-chain polymer aggregates, poor processability, and infusibility. Moreover, after a prolonged cycle time, its electrical conductivity also declines. Polyaniline has been combined with an additional adsorbent with more excellent adsorption capabilities, good regeneration ability, and selectivity utilizing straightforward synthesis procedures to develop novel nanocomposite adsorbents. The most attractive PANI/cellulose-based nanocomposites exhibit electrical conductivity because of the combined use of cellulose nanofillers and PANI matrix. Such nanomaterials, which have enhanced conductive, electrical, mechanical, and adsorbing capabilities, are widely used in the water treatment and electronics industries, as well as in the biomedical and electronics industries [24,26]. Both chemical and electrochemical oxidative polymerization in acidic media can be used to synthesize PANI. Ammonium persulfate (APS) and potassium persulfate (KPS) are the most often utilized initiators or oxidants for the chemical polymerization of aniline because they provide better conversion and yield. Several different macromolecular PANI structures exist due to the variance in experimental synthesis circumstances. In situ polymerization is the most popular technique for developing PANI nanocomposites [26]. Therefore, fabricating a WTP composite with polyaniline could be an excellent strategy to enhance pollutant adsorption ability and lower its wettability.

In this article, waste tissue paper and polyaniline (PANI@WTP) nanocomposite were synthesized via a facile single-step in situ oxidative polymerization of aniline in the presence of ammonium persulfate (NH_4_)_2_S_2_O_8_ as an oxidizing agent. The fabricated nanoporous polyaniline waste cellulosic tissue paper-based adsorbent was utilized for scavenging Cu(II) and phenol from synthetic wastewater solution. Variable adsorption parameters such as pollutant concentration, solution pH, and reaction time were investigated. Advanced adsorption kinetics and isotherm models were also fitted to the experimental data.

## 2. Materials and Methods

### 2.1. Chemicals

Ammonium persulfate extra pure (NH_4_)_2_S_2_O_8_) was purchased by SD Fine Chemicals India. Cupric Sulfate (CuSO_4_.5H_2_O), phenol (C_6_H_5_OH), and aniline were obtained from BHD Chemical Ltd., Poole, UK, and for the analysis of the copper solution, HACH Permachem reagents (HACH LANGE Gmbh, Duesseldorf, Germany) CuVer 1 copper reagent (category no. 2105869) was used. The DR/6000 UV-visible spectrophotometer was utilized to analyze the phenol as well as Cu(II) solutions.

### 2.2. Pulverization of WTP

The waste tissue paper used in the present study was collected from the industrial waste treatment laboratory of King Abd ulaziz University in Jeddah, Saudi Arabia, and sterilized in UV light for 2 h. The sterilized WTP was then washed with hot and cold water and heated in the oven drier for 24 h at 105 °C. Then, the dried WTP was soaked in deionized water for 30 min and exfoliated in a blender. This process was repeated several times until exfoliated WTP fibers were obtained. The exfoliated fibrous WTP was dried at 105 °C.

### 2.3. Synthesis of PANI@WTP Nanocomposite

The PANI@WTP composite was synthesized with aniline polymerization on the exfoliated WTP. Initially, 1 g of exfoliated WTP was suspended in 30 mL ethanol and sonicated for 10 min. After that, 150 mL 0.5 M HCl containing 1.35 mL aniline was added to the WTP suspended and stirred for half an hour in an ice bath. Ammonium persulfate (0.5 M) solution prepared in 1 M HCl (50 mL) was added dropwise for polymerizing the aniline onto WTP. The solution was agitated for 18 h. Blue-green PANI@WTP composite was then filtered and cleaned with water, ethanol, and acetone until the color was removed, then dried for 24 h at 60 °C in the oven. A similar method was used to synthesize polyaniline in the absence of WTP.

### 2.4. Characterization

To examine the surface structural and chemical compositions of waste tissue paper and the synthesized PANI@WTP composite, X-ray diffraction (XRD), Bruker D8 Advance X-ray diffractometer (Bruker Inc., Bremen, Germany) was used. A Fourier-transform infrared spectrometer (FTIR) model Agilent Cary 630 spectrometer was used to analyze the functional groups on WTP, PANI@WTP composite, PANI@WTP -Cu(II), and PANI@WTP-phenol before and after adsorption. The zeta potential was analyzed using Malvern panalytical serial no. MA1070389 in pH range of 2–10.

### 2.5. Adsorption of Cu(II) and Phenol

Batch-adsorption studies were completed using 0.02 g PANI@WTP composite dosage in 20 mL of Cu(II) and phenol solutions at various concentrations (10–800 mg/L), time (0–480 min), and solution pH (2–9). The adsorption tests were conducted in triplet, and the average results were reported. Reagents of 0.1 M HCl and 0.1 M NaOH were utilized to change the pH of the solution. After the adsorption of phenol onto the PANI@WTP composite, the residual concentrations of phenol in solutions were analyzed at wavelength λ-271 nm. CuVer 1 copper reagent (cat 2105869) was used to analyze the Cu(II) on the HACH DR-6000 spectrophotometer. The following equations determined the adsorption at equilibrium:(1)qe=(C0– Ce)vm 
where, *q_e_* is the uptake capacity of Cu(II) and phenol in (mg/g) onto PANI@WTP composite at equilibrium, *C_e_* is Cu(II) and phenol concentrations in solution (mg/L) at equilibrium, and C_0_, the initial Cu(II) ion and phenol solutions (mg/L). The volume (*v*) of the solution is in liters, and the dry weight (*m*) of PANI@WTP composite is in grams. 

### 2.6. Desorption and Regeneration Studies

The desorption of both pollutants and regeneration of the spent PANI@WTP composite adsorbent was carried out using 0.1 M NaOH and 0.1 HCl. A fixed amount (0.02 g) of spent PANI@WTP composite was mixed with 20 mL of 0.1 M NaOH or 0.1 HCl for pollutant desorption in the solution for 3 h under shaking conditions. Then, PANI@WTP was filtered and thoroughly washed with deionized water and dried at 70 °C for 16 h. The dried PANI@WTP was again used as an adsorbent to remove Cu(II) and phenol at the optimum adsorption conditions. A similar adsorption-desorption process was repeated for up to five cycles.

### 2.7. Synthetic Tap and Groundwater Purification

The adsorption performance of the PANI@WPT was also tested for the removal of the Cu(II) and phenol from the synthetic tap and groundwater. Therefore, a solution of Cu(II) and phenol was prepared by adding the appropriate amount to the tap and groundwater. A fixed amount of CuSO_4_.5H_2_O and phenol were added to prepare the 20 mg/L and 100 mg/L concentration solution in tap and groundwater. The adsorption studies were performed by mixing 0.02 g PANI@WPT in 20 mL prepared solution at 30 °C for 3 h at pH 5.2 for Cu((I) and pH 5 for phenol.

## 3. Results and Discussions

### 3.1. Synthesis and Characterization

Tissue paper (TP) is a flexible, soft, lightweight absorbent made of cellulose fibers. The paper pulp containing cellulose fiber is used to make the TP sheets. The WTP sheet was pulverized into a fluffy fibrous texture utilizing the blender. The idea to immobilize polyaniline onto the WTP was to reduce the hydrophilicity of the cellulose fibers and enhance the adsorption affinity toward pollutants. The existence of the hydroxyl functional groups on the cellulose makes the TP absorb more water. The interaction of the cellulose’s hydroxyl groups with the amine groups of the polyaniline can reduce the wettability of the TP and add more active sites for the interaction with the pollutants. The proposed scheme for the PANI@WTP synthesis is shown in Figure 1. To confirm successful synthesis, PANI@WTP was characterized by XRD, TEM, and FTIR spectroscopy.

The XRD pattern of the WTP and PANI@WTP is shown in Figure 2. The XRD spectrum of WTP displayed three major peaks at 2θ = 16.43°, 22.66°, and 34.47° corresponding to reflection lattice planes (110), (200), and (004) of the cellulose fiber structures. The highly crystalline structure of the WTP indicates the cellulose nanocrystal structure [25]. The XRD pattern of the PANI is shown in Appendix A, which shows two characteristic peaks centered at 19.35° and 25.65°. In comparison to the WTP and PANI, the XRD pattern of PANI@WTP nanocomposite showed a slight change in the peak positions (low) and peak intensity (increases) at 2θ = 15.29°, 22.33°, 31.57°, and 34.49°. The slight reduction in peak positions can be explained based on the polyaniline interaction with the WTP. The increase in peak height can be due to the solubilization of hemicellulose in acidic conditions during the PANI@WTP synthesis. The TEM images of WTP and PANI@WTP composite material are shown in Figure 3. The TEM image of WTP Figure 3a,b shows a fiber-like elongated network mainly due to the cellulosic structure. The TEM image of the PANI@WTP composite in Figure 3b shows the porous PANI particles deposited on the surface of WTP. The PANI particles are porous and of different shapes and sizes. However, the PANI particle alone shows various shapes and size particles, as shown in Appendix A. These results indicate the WTP has an effect on the porosity of the PANI in the PANI@WTP composite.

The FTIR spectrums of WTP and PANI@WTP before and after phenol and Cu(II) adsorption are displayed in Figure 4. The FTIR spectra of the PANI showing the characteristic peaks at 3439, 1575, 1475 1297, 1113, and 796 cm^−^¹ belong to the amine group, nitrogen quinine, benzene ring, C–N, and C=N stretching vibrations. The peak that appeared at nearly 3340 cm^−1^ from WTP was formed as a result of –OH stretching vibrations of the hydroxyl group on the cellulose chemical structure [27]. This peak shifted to 3323 cm^−1^ after aniline polymerization with WTP, indicating the interaction between the –OH group of cellulose and –NH group of PANI. The peak observed at 2900 cm^−1^ was due to the vibrational stretching of C−H group on the surface of the biopolymer [28]. The narrow peaks at 1428 cm^−1^ in the lower frequency region of WTP were due to the –CH_2_ group on the adsorbent. The N–H and –OH stretching vibration of the amino group of polyaniline and hydroxyl group of WTP were assigned to a characteristic peak at 3323 cm^−1^ in PANI@WTP composite [29]. However, the absorption peak observed at 3323 cm^−1^ was shifted to 3339 cm^−1^ in PANI@WTP–Cu(II) due to the complexation of the hydroxyl and amide group with Cu(II) ion after adsorption [29]. The aromatic compound (C–N) on the composite adsorbent formed a tiny peak at 1312 cm^−1^ [30]. The peak at 1024 cm^−1^ is attributed to the C–O–C ring in WTP [31]. This peak was shifted to 1026 cm^−1^ after adsorption with phenol. The shift of the peak band from 2900 cm^−1^ to 2809 cm^−1^ in PANI@WTP was attributed to the stretching vibration of the hydrocarbons (C–H) and hydrogen bonding between WTP and PANI during polymerization [27]. The same peak value (2809 cm^−1^) was also reported for polyaniline-impregnated nano cellulose adsorbent for the adsorption of hexavalent chromium [32]. The peak region between 1000 and 1150 cm^−1^ designated to the C–O–C bond vibration and this peak was like the peak band at 1158 cm^−1^ as illustrated in Figure 4. It is also noted that after adsorption with the metal ion Cu(II) and phenol, the adsorption peak corresponding to the phenolic –OH and NH^+^ functional groups changed. The change in wavenumber is an the implication of the participation of different functional group on Cu(II) and phenol adsorption from the aqueous media and the formation of bonds with PANI@WTP [33]. 

### 3.2. Scavenging of Cu(II) and Phenol

In the comparative adsorption tests for removing the Cu(II) and phenol from the aqueous solution onto WTP, PANI, and PANI@WTP nanocomposite, 0.02 g of material was mixed with a 20 mL (100 mg/L) solution at a pH of 5. The results (Appendix A) indicated that the adsorption efficiency of the Cu(II) and phenol onto WPT, PANI, and PANI@WTP nanocomposite was 46, 38, and 96.23 mg/g and 30, 38, and 74.3 mg/g, respectively. The PANI@WTP nanocomposite showed a higher adsorption of Cu(II) and phenol than WPT and PANI due to a large number of functional groups present on the surface of the PANI@WTP nanocomposite. These findings were used to evaluate the effectiveness of PANI@WTP nanocomposite for Cu(II) and phenol as a function of the adsorption parameters such as solution pH, pollutant concentration, and contact time.

### 3.3. Effect of Solution pH on Adsorption Process

The pH of a solution plays a significant role in determining the surface charge of the adsorbent during the adsorption mechanism and the extent of ionization of pollutants [34]. The zeta potential analysis of PANI@WTP nanocomposite at various pHs is shown in Figure 5a. As can be shown from Figure 5a, the point of zero charge (PZC) of the PANI@WTP nanocomposite is 9.76, which indicates that the adsorbent becomes positively charged at a pH < 9.76 and negatively charged at a pH > 9.76. The impact of the pH on the adsorption of Cu(II) and phenol was evaluated in the range of 2–5.5 and 2–10, respectively, The solution pH was adjusted between 2 and 5.5 for Cu(II) to avoid precipitation [35]. The highest scavenging of Cu(II) and phenol was achieved at pH 5.2 and 3, respectively. Cu(II) adsorption capability at the low pH range can be related to competition for similar adsorption sites between hydrogen ions and divalent Cu(II) ions [35,36]. The pH of the solution also has a significant influence on the phenol elimination as shown in Figure 5b. In the acidic media (pH 3–5), the removal of phenol was high. The high adsorption capacity at a lower pH for phenol might be due to the protonation of the organic functional groups such as carbonyl found both on the adsorbate and surface of the adsorbent and, as a result decreasing the electrostatic repulsive force [37]. A similar phenol adsorption has been reported by Fe–nano zeolite [38].

### 3.4. Effect of Contact Time on Adsorption and Kinetic Models

The Cu(II) and phenol adsorption onto the PANI@WTP composite was examined at various time intervals (0–420 min). The adsorption analysis was carried out using 20 mL of Cu(II) and phenol solutions with initial pollutant concentrations of 500 mg/L, pH 5.2 for Cu(II), and pH 5 for phenols at 30 °C, and the outcomes are shown in Figure 6. A quick removal of Cu(II) and phenol was detected during the first 5 min of the adsorption process, and subsequently, scavenging gradually increased [39,40]. It was discovered that the rate of phenol adsorption was slightly lower than Cu(II) solution. The higher adsorption of the Cu(II) onto PANI@WTP can be explained based on its smaller size than the phenol. The Cu(II) ionic radius is 0.73 A (0.074 nm), while phenol’s effective molecular diameter is 0.75 nm. The small-size Cu(II) can easily diffuse into the porous structure of the PANI@WTP [41,42]. Moreover, the fast removal of the Cu(II) ion at the beginning of the adsorption process might be due to the availability of more adsorption sites of PANI@WTP composite, and as time passed, the active sites were occupied [43].

The rate of the adsorption process for the uptake of Cu(II) and phenol was determined using adsorption kinetics. There are two main processes: physical adsorption and chemical adsorption. The physical adsorption is caused by weak attraction forces (van der Waals). In contrast, chemisorption necessitates the formation of a strong connection between the solvent and the substrate to permit the activation of atoms [44]. The rate of adsorption kinetics for Cu(II) and phenol in liquid–solid interactions has been modeled using pseudo-first-order, pseudo-second-order, and Elovich kinetic models [45]. The nonlinear pseudo-first, pseudo-second-order, and Elovich kinetic models are described in Equations (2)–(4), respectively.
(2)$$q_{t}=q_{e}\left(1-e^{k_{1}t}\right),$$
(3)$$q_{t}=\frac{k_{2}q_{e}{}^{2}t}{1+k_{2}q_{e}t}\;$$
(4)$$q_{t}=\frac{1}{\beta }+ln\left(\alpha \beta \right)+\frac{1}{\beta }ln\left(t\right)$$
where *q_e_* and *q_t_* are the amounts of adsorbate uptake per mass of adsorbent at equilibrium and at any time *t* (min), respectively, and k_1_ (min^−^¹) is the rate constant of the pseudo-first-order equation, and *k*_2_ (g mg^−1^ min^−1^), is the pseudo-second-order equation constant rate. Elovich constant α (mg g^−1^ min^−1^) is the adsorption rate, and β (mg g^−1^) is the desorption coefficient.

The nonlinear plots for applied kinetic models for Cu(II) adsorption and phenol onto PANI@WTP are shown in Figure 6a,b, respectively. The computed parameters of the kinetic models are illustrated in Table 1. The correlation coefficient (R^2^) values for pseudo-first-order kinetic, pseudo-second-order, and Elovich kinetic models are examined to assess the suitability of the experimental data, as tabulated in Table 1. The calculated kinetic parameter correlation coefficient (R^2^) value suggested the Elovich kinetic model had a high (R^2^) value (0.9485) for Cu(II) compared to the pseudo-first and pseudo-second-order kinetic model. The order of kinetic model best fitted the removal of Cu(II) by PANI@WTP was Elovich kinetic model > pseudo-second order > pseudo-first-order kinetic model, respectively. The scavenging of phenol onto PANI@WTP was best fitted to the pseudo-second-order kinetic model with a high R^2^ value (0.9746). The order of the best-suited kinetic model for phenol scavenging onto PANI@WTP was in the following order: pseudo-second-order kinetic > pseudo-first-order kinetic > Elovich kinetic model. Furthermore, the values of computed adsorption capacity q_e_cal.: 396.4154 mg/g for the scavenging of Cu(II) from the pseudo-second-order kinetic model was consistent to the data obtained from the experimental value q_e_exp: 397 mg/g, except for a negligible difference. The calculated adsorption capacity was q_e_cal.: 328.3705 mg/g for the phenol from the pseudo-second-order kinetic model was also close to the experimental data (320 mg/g).

### 3.5. Effect of Concentration on Adsorption and Isotherm Models

The adsorption capacity is significantly influenced by the initial Cu(II) and phenol concentrations [26]. The impact of the initial feed concentration of Cu(II) and phenol on adsorption is shown in Figure 7a,b, respectively. As seen from Figure 7, adsorption capacity increases from 8.86 mg/g to 397 mg/g for Cu (II) and 2.45 to 320 mg/g for phenol. The capacity of the PANI@WTP increases as the initial amount of Cu(II)/phenol rises. As the initial pollutant concentration rises, the number of Cu(II) and phenol-filled adsorptive sites of PANI@WTP increases, resulting in increased adsorption capacity. Cu(II) ions and phenol competed for a fixed number of dynamic locations on the adsorbent at greater concentrations. As a result, the contaminant molecules have insufficient binding sites on PANI@WTP at higher concentrations [46].

The well-known adsorption isotherms, namely the Langmuir, Freundlich, and Temkin adsorption isotherms, were applied to assess the interaction among PANI@WTP and adsorbate. According to the Langmuir adsorption concept, molecules are adsorbed at a definite number of well-defined catalyst surfaces, evenly distributed across the adsorbent’s surface. There is no connection between the adsorbate species because these binding sites have a similar potential for the adsorption of a monomolecular layer [31]. The Freundlich isotherm is useful when working with heterogeneous sorbent media to determine the sorption phenomenon. The Freundlich isotherm is assumed from the principle that adsorptive sites disperse exponentially in relation to the heat of the sorption model. The Temkin isotherm model describes the interactions of the adsorbent and adsorbate during adsorption progression. The assumptions of the Temkin model are based on the concept that during the adsorption of the sorbate and sorbent, neglecting the effect of extremely low and large values of concentrations, the heat of the adsorption will not be changed. Rather, the heat declines with coverage because of the interactions of the sorbate and sorbent during adsorption [47]. The following equations, respectively, represent the Langmuir, Freundlich, and Temkin isotherms.
(5)$$q_{e}=\frac{q_{m}K_{L}Ce}{1+K_{L}Ce}$$
(6)$$q_{e}=K_{F}Ce^{1/n}\;$$
(7)$$q_{e}=B_{t}\;\ln\left(K_{t}C_{e}\right)$$
where *K_F_* (the strength factor) and *n* are the Freundlich adsorption constants that were determined from the intercept and slope of the linear plots of log *q_e_* vs. log *Ce*, respectively and Langmuir constants, K_L_ (L/mg) are constants that are related to adsorption capacity and energy or net enthalpy of adsorption, respectively. *q_e_* is the corresponding adsorption capacity (mg g^−1^), q_m_ represents maximum adsorption capacity (mg g^−1^) and Ce is the Cu(II) and phenol equilibrium concentration (mg/L) respectively. *K_t_* and *B_t_* are the Temkin constants that refer to the heat of adsorption. The Langmuir, Freundlich, and Temkin isotherm models for the adsorption of Cu(II) and phenol onto PANI@WTP are illustrated in Figure 7a,b, respectively. The computed parameters of the Langmuir, Freundlich, and Temkin isotherm models computed by nonlinear regression of *q_e_* vs. Ce are tabulated in Table 2. According to the higher R^2^ values in Table 2, the Langmuir isotherm model (R^2^, 0.9439) followed by the Freundlich isotherm model (R^2^, 0.8699) best described the experimental results for the adsorption of Cu(II) onto PANI@WTP. However, the R^2^ value (0.9647) estimated for the adsorption of phenol on PANI@WTP nanocomposite, calculated from the Langmuir isotherm model equation, followed by the Temkin isotherm model, was well suited to the experimental results.

### 3.6. Cu(II) and Phenol Adsorption Mechanism

The scavenging Cu(II) and phenol onto the PANI@WTP nanocomposite was influenced by several parameters, which include physical and chemical features as well as the surface features of the adsorbent and the nature of the pollutant itself. The higher adsorption capacity of the PANI@WTP nanocomposite than the WTP was due to the large number of active sites. The WPT only has the cellulose-based active sites (–OH and =O), while after coupling with PANI, the PANI@WTP nanocomposite has –OH, =O, NH, =N, and NH_2_ adsorption sites for Cu(II) and phenol. The effective adsorption of Cu(II) ions onto PANI@WTP could be described as the interaction between Cu(II) ions and the hydroxyl and amine-functional groups [30]. The increase of adsorption capacity with increasing pH for Cu(II) ion suggested that the adsorption mechanism might be due to the ion exchange properties of positively charged Cu(II) and H^+^ on ^+^NH_2_ of the composite [31]. Moreover, as the solution pH increases, the surface charge of the adsorbent becomes more negative, which favours the adsorption of the Cu(II) at the higher pH. Furthermore, the adsorption mechanism of Cu(II) ions might be described based on the FTIR analysis. The change in wave number indicated the interaction of Cu(II) ion via complexation with –OH and amide functional groups on the surface of the material [35]. Moreover, the adsorption of phenol onto PANI@WTP may be due to the strong interactions of the NH**^+^** from the polyaniline and delocalized oxygen atoms from the phenol [48]. Furthermore, the pseudo-second-order kinetic model fitting for phenol adsorption onto PANI@WTP indicated that the phenol adsorption process might be due to the interactions of π-π electrons from the aromatic ring on phenol and PANI [49]. In general, the adsorption is a complex process with more mechanisms; hence, hydrogen bonding, physical adsorption, and π-π interactions might take part in the adsorption process of Cu(II) and phenol removal onto the PANI@WTP composite [50,51].

### 3.7. Regeneration

The regeneration of the used adsorbent material is essential to make the adsorption process cost effective [52]. The reusability of the PANI@WTP nanocomposite was studied for five cycles, as shown in Figure 8. The adsorption–desorption experimental results depicted that the PANI@WTP nanocomposite adsorbent can be regenerated several times with a slight loss in adsorption capability. This might be due to the deactivation of the active sites and incomplete desorption of the pollutant species from the inner structure of the adsorbent.

### 3.8. Comparison of the Adsorption Efficiency

The PANI@WTP nanocomposite efficiency was compared with other adsorbents utilized for Cu(II) and phenol adsorption. Table 3 shows the comparative adsorption capacity of the other adsorbents for Cu(II) and phenol from literature. The findings of the comparison demonstrate that the PANI@WTP nanocomposite is an effective material for scavenging Cu(II) and phenol from wastewater.

### 3.9. Synthetic Tap and Groundwater Purification

The adsorption performance of PANI@WTP was tested to remove Cu(II) and phenol from the tap and groundwater, and results are included in Figure 9. As seen in Figure 9a, PANI@WTP’s ability to adsorb the Cu(II) was approximately 14 mg/g for groundwater and 11.3 mg/g for tap water, while the phenol adsorption capacity was 5.19 mg/g and 5.14 mg/g at 20 mg/L. As shown in Figure 9b, a removal capacity of 43.23 mg/g and 21 mg/g was recorded for Cu(II) and 13.23 mg/g and 27 mg/g for the phenol adsorption onto the PANI@WTP composite at a 100 mg/L concentration for ground and tap water, respectively. The adsorption capacity of the adsorbent was also compared with deionized wastewater samples. It was clearly seen that the solution preparations in the tap, groundwater, or deionized water affect the adsorption process. The low adsorption of the pollutants from tap and groundwater compared to deionized water might be related to ionic species such as minerals, anions, halides etc., in the ground and tap water, which hindered the treatment process [66].

## 4. Conclusions

In this article, waste tissue paper and a polyaniline (PANI@WTP) nanocomposite were synthesized using a single-step in situ oxidative polymerization of aniline in the presence of ammonium persulfate (NH_4_)_2_S_2_O_8_ as an oxidizing agent. The fabricated PANI@WTP composite was examined for the decontamination of Cu(II) and phenol from wastewater. It has been found that the optimum pH values for both Cu(II) and phenol adsorption onto PANI@WTP composite were 5.2 and 3, respectively. The Elovich kinetic model and pseudo-second-order kinetic model fitted the experimental kinetic data for Cu(II) and phenol adsorption, respectively, whereas the Langmuir equilibrium isotherm model was well fitted for adsorption of both Cu(II) and phenol onto the PANI@WTP composite. In general, the study results suggested that the fabricated composite material is recyclable and still a promising candidate adsorbent for treating pollutants from wastewater.

## Figures and Tables

**Figure 1 nanomaterials-13-01014-f001:**
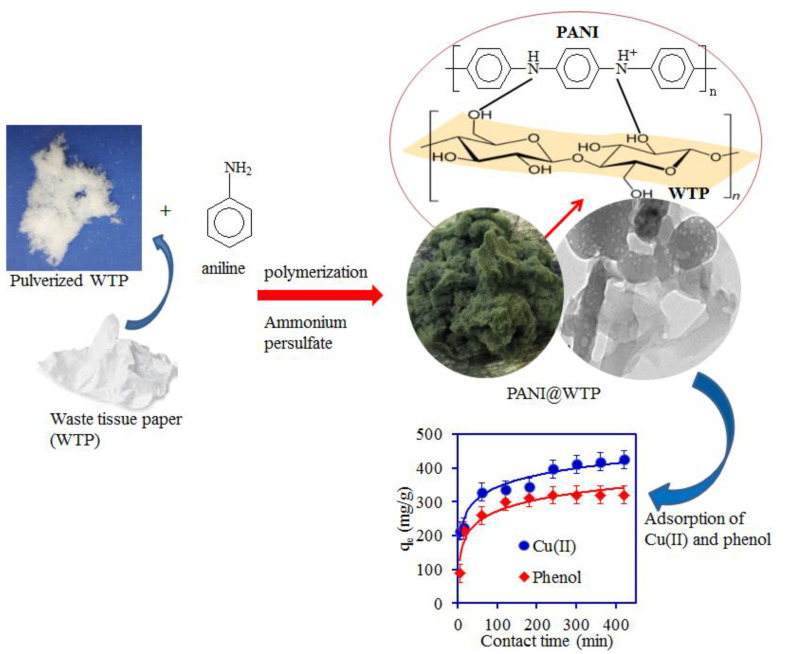
A proposed scheme for the PANI@WTP nanocomposite synthesis and application.

**Figure 2 nanomaterials-13-01014-f002:**
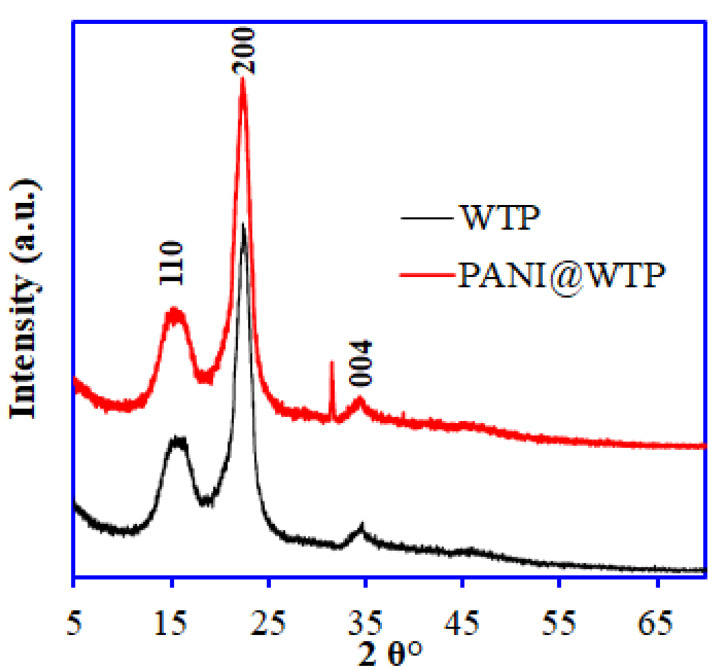
XRD pattern of WTP and PANI@WTP nanocomposite.

**Figure 3 nanomaterials-13-01014-f003:**
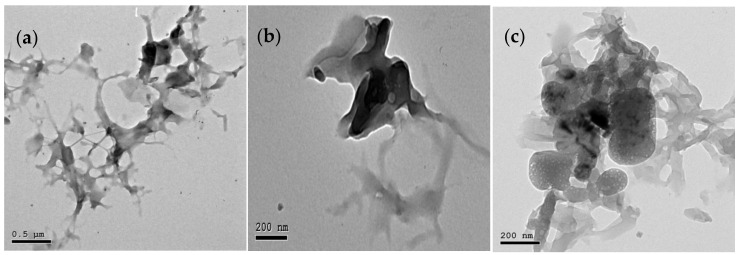
TEM images of (**a**,**b**) WTP and (**c**) PANI@WTP nanocomposite.

**Figure 4 nanomaterials-13-01014-f004:**
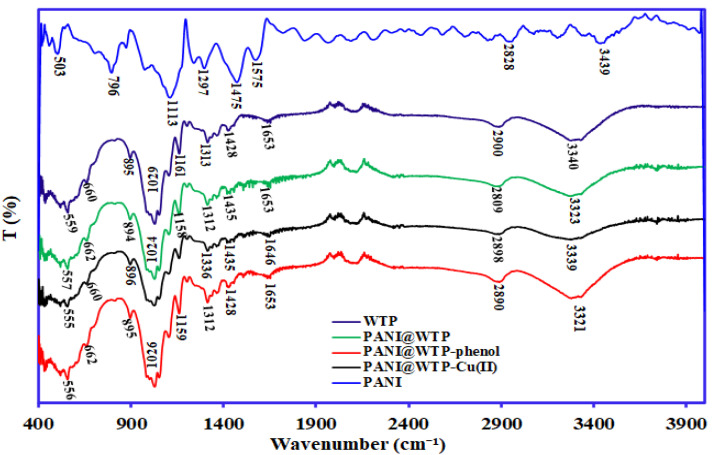
FTIR spectrum of WTP, PANI@WTP nanocomposite before and after pollutant adsorption.

**Figure 5 nanomaterials-13-01014-f005:**
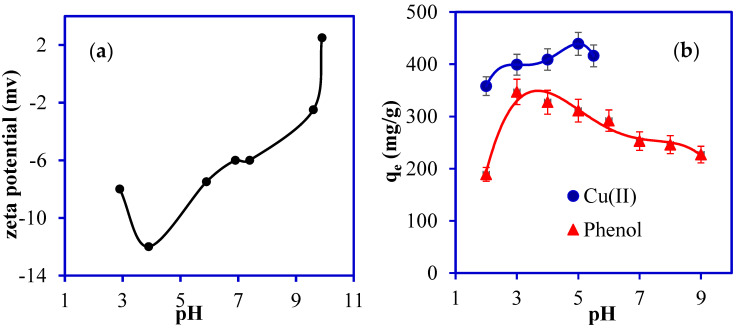
(**a**) Zeta potential analysis, (**b**) the effect of solution pH on adsorption of Cu(II) and phenol (concentration, 500 mg/L; adsorbent dosage, 0.02 g; volume, 20 mL; temp, 30 °C).

**Figure 6 nanomaterials-13-01014-f006:**
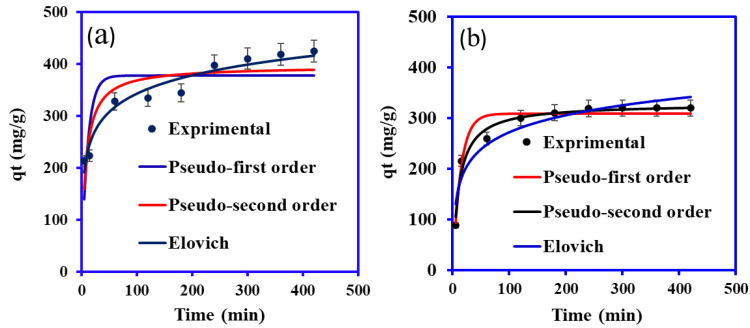
The kinetic plots for the adsorption of (**a**) Cu(II) and (**b**) phenol onto the PANI@WTP nanocomposite (concentration, 500 mg/L; adsorbent dosage, 0.02 g; volume, 20 mL; pH 5.2 for Cu(II), pH, 5 for phenol; temp, 30 °C).

**Figure 7 nanomaterials-13-01014-f007:**
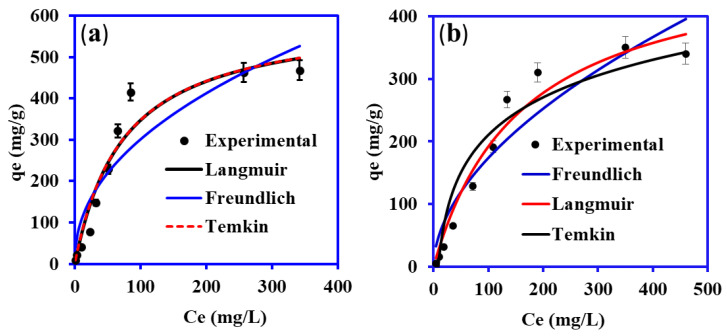
Adsorption isotherm plots for (**a**) Cu(II) and (**b**) phenol onto PANI@WTP composite (adsorbent dosage, 0.02 g; volume, 20 mL; contact time, 5 h; temp, 30 °C; pH 5.2 for Cu(II), pH 5 for phenol solution).

**Figure 8 nanomaterials-13-01014-f008:**
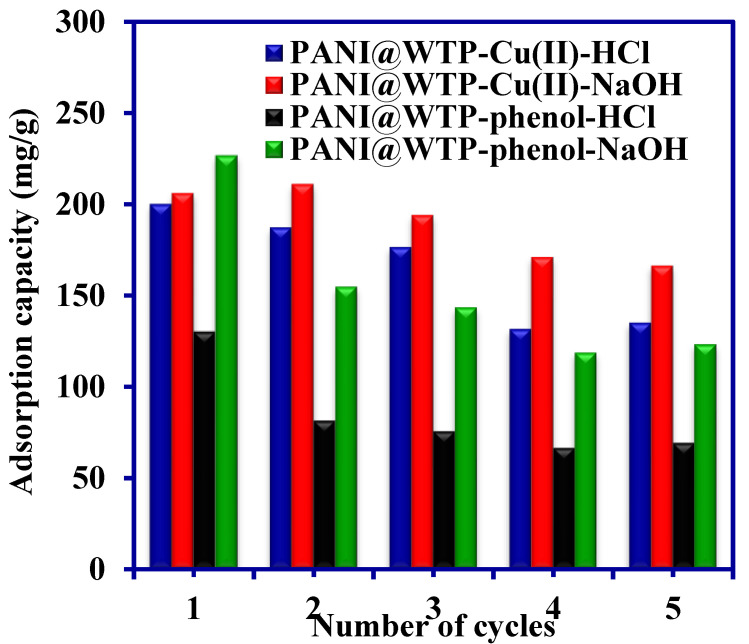
Regeneration of PANI@WTP nanocomposite (concentration, 500 mg/L; volume, 20 mL; adsorbent mass, 0.02 g; pH 5 for phenol, 5.2 for Cu(II); temp, 30 °C).

**Figure 9 nanomaterials-13-01014-f009:**
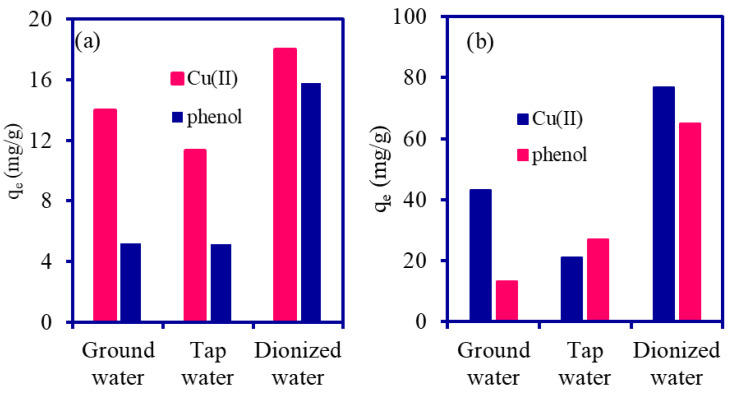
Adsorption of Cu(II) and phenol from synthetic tap and groundwater using PANI@WTP (**a**) 20 mg/L and (**b**) 100 mg/L (volume, 20 mL; adsorbent mass, 0.02 g; pH 5 for phenol, 5.2 for Cu(II); temp, 30 °C).

**Table 1 nanomaterials-13-01014-t001:** Adsorption kinetics parameters for Cu(II) and Phenol onto PANI@WTP nanocomposite.

Kinetic Model	Parameters	Cu(II)	Phenol
**Pseudo-first-order**	**q_e_(exp) (mg g^−1^):**	397	320
	**q_e_(cal) (mg g^−1^):**	377.499	308.525
	***k*_1_(min^−1^):**	0.0920	0.0726.
	**R^2^:**	0.6195	0.9431
**Pseudo-second-order**	**q_e_(cal) (mg g^−1^):**	396.41	328.370
	***k*_2_(g mg^−1^ min^−1^):**	0.339 × 10^−3^	0. 285 × 10^−3^
	**R^2^:**	0.7883	0.9746
**Elovich model**	***a*(mg g^−1^ min^−1^):**	425.615	132.691
	***β*(mg g^−1^):**	0.0195	0.0206
	**R^2^:**	0.9485	0.9121

**Table 2 nanomaterials-13-01014-t002:** Adsorption isotherm parameters for Cu(II) and phenol onto PANI@WTP nanocomposite.

Isotherm Model	Parameters	Cu(II)	Phenol
**Langmuir**	**q_m_(mg g^−1^):**	605.204	501.234
	**K_L_(L mg^−1^):**	0.0135	0.0062
	**R^2^:**	0.9436	0.9647
**Freundlich**	**n:**	2.2012	1.8580
	**Kf(mg g^−1^) (mg L^−1^)^−1/n^:**	37.152	14.591
	**R^2^:**	0.8699	0.9061
**Temkin**	**B_t_(J mg^−1^)**	27.203	29.083
	**K_t_(L mg^−1^):**	0.3590	0.1136
	**R^2^:**	0.8411	0.9113

**Table 3 nanomaterials-13-01014-t003:** Comparison of maximum adsorption capacity of Cu(II) and phenol with other materials.

	Experimental Conditions					
Adsorbate	Adsorbents	pH	Conc. (mg/L)	Contact Time (min)	q_e_(mg g^−1^)	Ref.
Cu(II)	Musk melon	7	50	120	78.74	[53]
	Banana peel	6	-	1440	28.57	[54]
	Sugarcane bagasse	5	10	60	88.9	[55]
	Fly ash	5	500	-	207.3	[56]
	Wheat bran	5	100	30	51.5	[57]
	Cellulose pulp waste	6	100	180	4.98	[58]
	Orange peel	5	100	180	289	[59]
	Carbon	5	100	145	77-83	[60]
	PANI@WTP	5.2	500	240	605.20	**This study**
Phenol	Banana peel	7	30	180	688.9	[61]
	Biochar	6.5	50	270	26.738	[62]
	Tea waste biomass	7	50	1440	9.487	[63]
	Lead ferrite-MAC	2	250	60	158.9	[64]
	Ziziphus leaves	6	20	240	15	[65]
	PANI@WTP	5	500	180	501.23	**This study**

## Data Availability

Data is available on request.

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
