# Peer review of "Facile Synthesis of the Polyaniline@Waste Cellulosic Nanocomposite for the Efficient Decontamination of Copper(II) and Phenol from Wastewater"

_nanomaterials, 2023, doi:10.3390/nano13061014_

Round 1

Reviewer 1 Report

This study reported the synthesis of polyaniline-embedded waste tissue paper (PANI@WTP) as a removal of copper(II) and phenol from aqueous solution. 

There are several comments should be addressed as follows:

1- In the Abstract part:

The abstract part needs to be revised and modified to present only the most relevant goal, idea, and get quickly to the point of the paper. 

Moreover, the merits and novelty of the current study need further clarifications.

2- Regarding the introduction section:

In the introduction part, the author should highlight and clarify the following:

a- The author should clarify how the current research problem has gotten worse, as well as the existing challenges, and the available solutions

b- It is necessary to highlight the different techniques used in the removal of targeted species, with an explanation of their advantages and disadvantages. Then, the author should shed light on the reason and the importance of the technique used

c- More consideration ought to be given to the design of the materials utilized and the fieldwork initiatives connected to the current investigation. 

in addition, emphasizing the uniqueness of the current design.

d- A clear claim should be provided at the end of the introduction. Thus, the claim part presented at the end of this introduction should be revised with highlighting the objective, adsorbent design, technology used, innovation and key findings.

e- There are some related citations that can help authors as follows:

Journal of Hazardous Materials 406, (2021) 124314 https://doi.org/10.1016/j.jhazmat.2020.124314   &     Separation and Purification Technology 116, (2013)  73-86   https://doi.org/10.1016/j.seppur.2013.05.011              &            Analyst 139 (24), (2014) 6393-6405    https://doi.org/10.1039/C4AN00980K

Note: the suggested relevant references to clarify comments raised and assist authors, moreover, use them in this work depends on their relevance and the author's vision 

3- In the experimental part:

In general, the synthesis of targeted adsorbent needs to be carefully revised with more clarification.

 In this regards, it is better to add a clear schematic diagram showing all successive preparation steps highlighting the essential condition to achieve each step, some analysis may be added to clarify the output. In addition, it is preferable to add the applicable side (measurement) and testing design.

** Therefore, figure 1 must be redesign based on this comment

4- In the results and discussion section:

The Results and discussion part needs to be carefully revised as follows

a- In general, the results and discussion part is short and lacks a clear discussion of the materials and results obtained. Thus, the author should provide further interpretation and debate about the results and the data obtained to help the readers

b- The rules and mechanism of materials formation needs further discussion and clarification. 

c- The analysis of PANI should be presented in addition to the WTP and PANI@WTP 

d- In figure 2, the TP is present inside the figure, and WTP in the figure's caption

e- TEM has been presented in a different scales "Figure 3. TEM images of (a) WTP and (b) PANI@WTP nanocomposite."

Thus it is recommended to use the same scale. In addition the analysis of PANI should be added

f- Is the final product a homogeneous composite?

g- In Figure 5,

    i- why did the author use a different pH range with phenol and copper ions?

   ii- What is the effect of raising the pH with copper ions?

   iii- Moreover, is this material stable at higher pH?

h- What is the active part in this process?

** Is there any effect of WTP on the removal process?

** The author has to test the removal performance process using PANI and WTP as well as the compound.

i- The mechanism of this process needs further clarifications, thus the part of "3.3. Cu(II) and phenol adsorption mechanism" should be revised

   it is well-known that pH affecting the doping of PANI and change its form

j- Is there any effect of using different anions of copper?

k- The effect of coexistence ions (competitive ions) should be tested and discussed to confirm the applicability.

l- The real sample should be tested

m-  The figures provided should be redesign with high precision.

n- Moreover, the figure captions is not enough to describe the figure, the author must provide more information and details enough to clarify the figures.

Author Response

Dear Reviewer

Thank you for your valuable comments and suggestions. The whole manuscript has been revised according to your comments and suggestion and changes are highlighted in blue.

Response to Reviewer comments

This study reported the synthesis of polyaniline-embedded waste tissue paper (PANI@WTP) as a removal of copper(II) and phenol from aqueous solution. 

There are several comments should be addressed as follows:

  1. In the Abstract part:

The abstract part needs to be revised and modified to present only the most relevant goal, idea, and get quickly to the point of the paper. 

Moreover, the merits and novelty of the current study need further clarifications.

Response: The abstract has been revised.

 2- Regarding the introduction section:

In the introduction part, the author should highlight and clarify the following:

  • The author should clarify how the current research problem has gotten worse, as well as the existing challenges, and the available solutions

Response: Thanks. The research problem has been revised in the first part of the introduction.  

  • It is necessary to highlight the different techniques used in the removal of targeted species, with an explanation of their advantages and disadvantages. Then, the author should shed light on the reason and the importance of the technique used

Response: Thank you for the suggestion. Different techniques applied to remove heavy metals are cited, and the disadvantages of those technologies have discussed the merits of the applied technology.

c- More consideration ought to be given to the design of the materials utilized and the fieldwork initiatives connected to the current investigation. In addition, emphasizing the uniqueness of the current design.

Response: Thanks. The section has been revised and more explanation has been added.

d- A clear claim should be provided at the end of the introduction. Thus, the claim part presented at the end of this introduction should be revised with highlighting the objective, adsorbent design, technology used, innovation and key findings.

Response: Thank you for your valuable comment. The objective, adsorbent design and method used to synthesize the adsorbent are all stated and corrected at the end of introduction.

e- There are some related citations that can help authors as follows:

Journal of Hazardous Materials 406, (2021) 124314 https://doi.org/10.1016/j.jhazmat.2020.124314   &     Separation and Purification Technology 116, (2013)  73-86   https://doi.org/10.1016/j.seppur.2013.05.011              &            Analyst 139 (24), (2014) 6393-6405    https://doi.org/10.1039/C4AN00980K

Note: the suggested relevant references to clarify comments raised and assist authors, moreover, use them in this work depends on their relevance and the author's vision 

 Response: Thank you for sharing these valuable references. The All the refrences has been included in the manuscript.

3- In the experimental part:

In general, the synthesis of targeted adsorbent needs to be carefully revised with more clarification.

 In this regards, it is better to add a clear schematic diagram showing all successive preparation steps highlighting the essential condition to achieve each step, some analysis may be added to clarify the output. In addition, it is preferable to add the applicable side (measurement) and testing design.

** Therefore, figure 1 must be redesign based on this comment

Response: Thank you for your valuable suggestion. Figure 1 has been revised according to suggestion.

4- In the results and discussion section:

The Results and discussion part needs to be carefully revised as follows

  • In general, the results and discussion part is short and lacks a clear discussion of the materials and results obtained. Thus, the author should provide further interpretation and debate about the results and the data obtained to help the readers

Response: The whole manuscript has been revised and more explanations have been added and highlighted

  • The rules and mechanism of materials formation needs further discussion and clarification. 

Response:  The adsorption mechanism is modified and highlighted in the manuscript.

  • The analysis of PANI should be presented in addition to the WTP and PANI@WTP 

Response: XRD spectrum and SEM image of the PANI have been included in the supplementary file and a related description has been included in the text.  

  • In figure 2, the TP is present inside the figure, and WTP in the figure's caption

Response: Thank you. Corrected as WTP.

  • TEM has been presented in a different scales "Figure 3. TEM images of (a) WTP and (b) PANI@WTP nanocomposite."

Thus it is recommended to use the same scale. In addition, the analysis of PANI should be added

Response:  Thanks, Same magnified image has been included. XRD and SEM image of the PANI has been included in the supplementary file

  • Is the final product a homogeneous composite?

Response:   TEM image showed that the final product is not homogeneous. 

g- In Figure 5,

  • why did the author use a different pH range with phenol and copper ions?

Response: The solution pH was adjusted between 2 to 5.5 for Cu(II) to avoid precipitation.  Phenol does not cause such type of behavior; therefore wide pH range was studied for phenol.

  • What is the effect of raising the pH with copper ions?

Response: We appreciate your important comment. Increasing the pH enhanced the adsorption capacity of the adsorbent from our experimental observation. However, we have noted that pH above 6 for copper ions forms precipitates in form of Cu(II) hydroxides.

  • Moreover, is this material stable at higher pH?

Response: PANI@WTP nanocomposite particles have a point of zero charge (PZC) at a pH of 9.76. These results suggest that the prepared material is stable.

h- What is the active part in this process?

Response: The active sites on the cellulose and polyaniline are the main active sites on the adsorbent. 

** Is there any effect of WTP on the removal process?

Response: WTP is also a promising adsorbent. We included the adsorption capacity of the WTP in section 3.2 and Fig. S3.

** The author has to test the removal performance process using PANI and WTP as well as the compound.

Response: The adsorption efficiency of the PANI and WTP to remove Cu(II) and phenol were evaluated, and the results are included in Fig 3 and section 3.2. It was observed that the composite gave good adsorption results.

i- The mechanism of this process needs further clarifications, thus the part of "3.3. Cu(II) and phenol adsorption mechanism" should be revised

   it is well-known that pH affecting the doping of PANI and change its form

Response: Thank you for highlighting your concern. Adsorption mechanisms are revised in the manuscript and highlighted

j- Is there any effect of using different anions of copper?

Response: We are working on the simultaneous adsorption of Cu(II) and phenol for the solution using the  PANI@WTP. In the new manuscript, we are studying the effect of different ions on the adsorption process along with the real sample analysis. Column adsorption studies would be part of the new study. Hopefully, we will submit the new manuscript soon.

k- The effect of coexistence ions (competitive ions) should be tested and discussed to confirm the applicability.

Response: The preliminary results revealed that coexisted ions affect the adsorption process. These results will be included in the new manuscript.

l- The real sample should be tested

Response: 3.9. Synthetic tap and groundwater purification analysis has been included. Real sample results will be included in the new manuscript.

m- The figures provided should be redesign with high precision.

Response: Thank you. Better-quality Figures are included.

n- Moreover, the figure captions is not enough to describe the figure, the author must provide more information and details enough to clarify the figures.

Response: The Figure captions with experimental details are provided and corrected in the manuscript. More information on the experiment has been added in the material and method section.

Reviewer 2 Report

Recommendation: Major

A. N. Doyo et al. had utilized waste tissue paper to synthesize polyaniline-embedded waste tissue paper (PANI@WTP) to remove Cu(II) and phenol. Materials were well characterized. The removal performance was studied comprehensively, including pH, kinetic model, etc. The mechanism is not very clear. Major revision is needed as following:

1.        The adsorption mechanism of PANI@WTP is not very clear. Why this material is much higher than PANI and WTP? With same weight, why composite has higher active sites?

2.        The surface area of materials is recommended to be done using N2 sorption isotherm.

3.        Figures are not clear.

4.        The shift of IR peaks should be marked in Figures.

5.        WTP should be used in Figure 2.

6.        Some sentences are incompetent, such as “Zeta potential of PANI@WTP was observed at pH 9.76.” in Line 197, Page 7.  “These results suggest that the adsorption of Cu(II) and phenol by PANI@WTP were formed mainly due to the.” in Line 263.

7.        Typo errors, such as “physisorption is caused…” in Line 230, Page, 8.

8.        The description of phenol removal model (Line 254-257) is not consisted with Figure 6b.

9.        Different concentration of pollute is corresponding to different qe. Therefore, some data in Table 3 is not suitable. For example, Fly ash concentration 50-150, qe 64. Orange peel…

10.     The regeneration process should be provided in the experimental part in detail. Washed by NaOH and HCl? Unclear.

Author Response

Dear Reviewer

Thank you for your valuable suggestions. The whole manuscript has been revised and the changes are highlighted in the manuscript.

Response to Review Reviewer  comments

  1. N. Doyo et al. had utilized waste tissue paper to synthesize polyaniline-embedded waste tissue paper (PANI@WTP) to remove Cu(II) and phenol. Materials were well characterized. The removal performance was studied comprehensively, including pH, kinetic model, etc. The mechanism is not very clear. Major revision is needed as following:

 The adsorption mechanism of PANI@WTP is not very clear. Why this material is much higher adsorption than PANI and WTP? With same weight, why composite have higher active sites?

Response: Thank you for your valuable suggestion. The adsorption mechanism has been explained based on a large number of the active sites on PANI@WTP than WTP, besides the other factors and adsorption forces. Please check the revised section.

  1. The surface area of materials is recommended to be done using N2 sorption isotherm.

Response: Surface area is an important factor in the adsorption process. We do not have this facility in our institution. We asked other centers for Surface area analysis, and they cannot do it in a short time since they are occupied.

  1. Figures are not clear.

Response: The figures are replaced with high-quality images.

  1. The shift of IR peaks should be marked in Figures.

Response: All the peaks are mentioned in the figure and discussed in the manuscript.

  1. WTP should be used in Figure 2.

Response: Corrected.

  1. Some sentences are incompetent, such as “Zeta potential of PANI@WTP was observed at pH 9.76.” in Line 197, Page 7.  “These results suggest that the adsorption of Cu(II) and phenol by PANI@WTP were formed mainly due to the.” in Line 263.

Response: The whole manuscript has been revised for language corrections. 

  1. Typo errors, such as “physisorption is caused…” in Line 230, Page, 8.

Response: Thanks for the observation. The whole manuscript has been revised carefully.  “physisorption”  corrected as physical adsorption

  1. The description of phenol removal model (Line 254-257) is not consisted with Figure 6b.

Response: Thanks for the observation. Before, Fig 6a and 6 b were the same, and it was a mistake. We replace Fig 6b. Now the Fig 6b and the text explain the same conclusion.

  1. Different concentration of pollute is corresponding to different qe. Therefore, some data in Table 3 is not suitable. For example, Fly ash concentration 50-150, qe 64. Orange peel…

Response: Table 3 has been revised, and parameter values are corrected.

  1. The regeneration process should be provided in the experimental part in detail. Washed by NaOH and HCl? Unclear.

Response:  A separate section has been added in the materials and method section as follows:  2.6. Desorption and regeneration studies. The desorption of both pollutants and regeneration of spent PANI@WTP composite adsorbent was done using 0.1M NaOH and 0.1 HCl. A fixed amount (0.02g) of spent PANI@WTP composite was mixed with 20 mL of 0.1M NaOH or 0.1 HCl for pollutant desorption in the solution for 3 hours under shaking conditions. Then, PANI@WTP was filtered and thoroughly washed with deionized water and dried at 70 °C for 16h. The dried PANI@WTP was again used as an adsorbent for the removal of Cu(II) and phenol at the optimum adsorption conditions. A similar adsorption-desorption process was repeated for up to five cycles.

Round 2

Reviewer 1 Report

The revised manuscript shows the efforts made by the author to cover most of the comments raised. but there are many comments should be addressed as follows:

1- The caption of figure 3 should be revised and corrected, (c) is missed. moreover, it is recommended to add further information.

2- Analysis of PANI is recommended to be added within figure 4

3- phenol should be added inside pH data in figure 5(b). In addition, why the same pH range didn't be used  for both targets?

4- The author has tested the Cu(II) target in high conc. 500 mg/L (500ppm), what about low concentrations?

5- What about removal of Cu(II) lower than permitted level?

Author Response

Dear Reviewer,

Thanks for your valuable comments and suggestions. The manuscript has been revised and the changes are highlighted in the manuscript.  

The revised manuscript shows the efforts made by the author to cover most of the comments raised. but there are many comments should be addressed as follows:

  1. The caption of figure 3 should be revised and corrected, (c) is missed. moreover, it is recommended to add further information.

Response: Thanks for the observation. Fig. 3 caption has been corrected as follows “Figure 3. TEM images of (a,b) WTP and (c) PANI@WTP nanocomposite”

2- Analysis of PANI is recommended to be added within figure 4.

Response: Thanks for the suggestion, FTIR spectrum of PANI has been added to Fig. 4 and a related justification has been added.

3- phenol should be added inside pH data in figure 5(b). In addition, why the same pH range didn't be used for both targets?

Response: Phenol data has indicated in Fig. 5b.  

We did not check the effect of pH for both pollutants at the same pH range because Cu(II) precipitation occurs beyond pH 5.5. Our focus was on adsorption experiments, not precipitation. Phenol does not show any structural change at the studded pH range. Therefore, we studied the effect of pH at different pH ranges for both pollutants.

4- The author has tested the Cu(II) target in high conc. 500 mg/L (500ppm), what about low concentrations?

Response: The saturation of all active sites of the adsorbent surface is very important to find the adsorption capacity at equilibrium concentration. Therefore, we used equilibrium concentration. We also check the adsorption of both pollutants at low concentrations and results are depicted in section 3.5 Effect of concentration on adsorption and isotherm models.

 5- What about removal of Cu(II) lower than permitted level?

Response: Dear Reviewer, As I mention in the first revision,  we are exploring this work on the real sample and performing column studies. We have collected groundwater and sludge samples which have multiple metals along with copper. The results will be included in the next work.

Reviewer 2 Report

well revised and can be accepted.

Author Response

Dear Reviewer

Thanks for considering our manuscript for publication.